# Do Negative Self-Evaluative Emotions Enhance Healthier Food Choices? Exploring the Moderating Role of Self-Affirmation

**DOI:** 10.3390/bs14070538

**Published:** 2024-06-26

**Authors:** Jingwen Li, Yu Chen, Mingcong Tang, Shuangmiao Wang

**Affiliations:** 1Faculty of Behavioural and Social Sciences, University of Groningen, 9712 TS Groningen, The Netherlands; jingwen.li@rug.nl; 2Faculty of Economics and Business, University of Groningen, 9747 AT Groningen, The Netherlands; y.chen@rug.nl; 3Department of Psychological and Brain Sciences, Boston University, Boston, MA 02215, USA; mingcong@bu.edu; 4Institute of Health Promotion and Medical Communication Science, Guangdong Medical University, Zhanjiang 524023, China

**Keywords:** healthy food consumption, negative self-evaluative emotions, self-affirmation, health behavioral change

## Abstract

Negative self-evaluative emotions arise when an individual engages in behavior that is perceived as inadequate or inconsistent with personal or societal norms and values, leading to feelings of inadequacy, shame, and dissatisfaction with oneself. These emotions are a central motivating force for changing unhealthy behaviors. However, negative evaluative emotions may also direct individuals towards defensive reactions such as reactance and avoidance. This can cause negative self-evaluative emotions to be less effective in reducing unhealthy behavior. More importantly, empirical evidence is needed to explore strategies for enhancing the effectiveness of interventions. In this study, we used an online experiment with 100 student participants to examine if increasing self-affirmation can increase the effectiveness of negative self-evaluative emotions in reducing unhealthy food consumption. We found that negative self-evaluative emotions can significantly increase healthy food consumption. However, our analysis did not reveal a significant moderating impact of self-affirmation on the effectiveness of negative self-evaluative emotions in reducing unhealthy consumption. This is the first study to explore the moderating impact of self-affirmation on the effectiveness of negative self-evaluative emotions on health behavioral change, which opens new avenues for studying how to apply the combination of stimulating negative self-evaluative emotions and increasing self-affirmation to induce behavioral change regarding healthy diets and even a broader range of fields.

## 1. Introduction

Previous studies demonstrate that healthy food choices (such as appropriate calorie intake) are the main determinants of preventing chronic diseases [1,2]. In contrast, unhealthy food choices (such as excessive meat consumption) have a high possibility of causing non-communicable diseases such as cardiovascular diseases and diabetes, which account for 63% of all deaths worldwide and will cost an estimated USD 30 trillion in the near future [3,4]. Therefore, to reduce the heavy burden of chronic diseases, population-wide interventions for decreasing unhealthy food consumption are recommended [5]. Economic incentives such as food taxation and nutritional education are the most common traditional interventions for promoting healthy diets [6,7]. Nevertheless, these tools have some unavoidable limitations, including high economic burdens, the difficulty of attracting the public’s attention, and so on [4,8].

One potential reason for the lack of effectiveness of common attempts at changing unhealthy food choices is the low motivation to change, which was implied to be much more important than increasing health knowledge [9]. The motivation behind food choices can be impacted by various emotional and cognitive factors. One modifiable and contributing factor is self-evaluative emotions, which are the self-conscious emotions that rely on how we assess ourselves and our actions with respect to internalized or imposed standards and norms [10]. These emotions are often influenced by social comparison or feedback from others, and they can impact an individual’s self-esteem, self-worth, and overall emotional well-being. Typical self-evaluative emotions encompass both positive aspects, such as pride, satisfaction, and a sense of accomplishment, and negative aspects, including guilt, shame, and embarrassment. Self-evaluative emotions often occur when individuals perceive the outcomes of a particular behavior [11]. Improper or dissatisfactory behavior, such as unhealthy food choices, can make individuals feel bad about themselves and stimulate negative self-evaluative emotions. Specifically, the more negative the consequences are perceived by the person who engages in the unhealthy behavior, the stronger the negative self-evaluative emotions will be [11]. The aversiveness of current negative self-evaluative feelings drives the development of expectations for relief and the emergence of positive self-evaluative emotions if unhealthy behavior is avoided [11]. Thus, negative self-evaluative emotions can warn people about harmful practices and encourage them to change their attitudes and behaviors [11,12].

Researchers have suggested using negative self-evaluative emotions, such as stress, comfort, and regret, as motivators for behavior change [11]. In empirical experiments, negative self-evaluative emotions were usually stimulated via health risk messages. The health risk contents would trigger a negative evaluation of the specific behavior, make people more aware of the danger they encounter, or increase their perception of personal risk, thus prompting them to change their unhealthy behaviors [13]. However, most research examining the relationship between negative self-evaluative emotions and health-related behavior has been conducted predominantly among individuals with mental health issues, such as addiction or emotional disorders [14]. The evidence for the intervention effectiveness of negative self-evaluative emotions for food choice among general populations remains scarce. Therefore, one primary aim of this study was to investigate whether negative self-evaluative emotions can indeed promote healthy food choices among the general population. This extension is crucial, particularly in light of the importance of population-based interventions aimed at reducing unhealthy food choices. By focusing on the general population, we aim to contribute to a better understanding of how interventions targeting negative self-evaluative emotions can be utilized in broader public health efforts.

Nevertheless, feeling bad about the unfavorable consequences may not always lead people to health behavior changes, and even though there is easily available knowledge about health hazards, recipients still often engage in harmful health behaviors [15]. This might be because individuals experiencing negative self-evaluative emotions often perceive that they have insufficient resources to handle the demands of their environment [16]. They would not perceive that they were able to effectively make changes. Another potential reason is that engaging in behavior with negative physical consequences is regarded as a threat to oneself because it makes one appear inadequate and non-adaptive [11]. Such health risk information implies we have not been adaptive or wise. This may violate our health values and pose a threat to our self-concept, which would elicit defensiveness to reduce this threat. As a result, message recipients may alter their interpretation of a behavior or avoid facing the threat to protect perceived self-worth [12]. For example, when individuals experience negative emotions, they may turn to food as a way to distract or numb themselves from the emotions. Therefore, it is important not only to investigate whether psychologically threatened people are likely to make health behavior changes [17], but also to understand how to maintain their motivation, particularly if they exhibit defensive behaviors.

Previous research suggested that the relationship between negative self-evaluative emotions and healthy behavior change (e.g., healthy food choices) may depend upon the perceived ability to effectively manage those negative self-evaluative emotions (hereafter referred to as self-affirmation) [17]. Self-affirmation was conceptualized as a procedure to reflect on core personal values, important personal strengths or attributes, and cherished social relations [18]. The purpose of self-affirmation is to enhance self-esteem and reduce the negative impact of threats to one’s sense of self. Research has shown that self-affirmation can have a range of positive effects, such as improving emotional well-being, reducing defensiveness and health information avoidance, offsetting threats to self-integrity and self-regulation of human behavior, and increasing motivation to engage in healthy behaviors. It can also be used as a coping mechanism in response to stress or negative feedback [19,20,21]. For example, Dijkstra [11] stated that when people are confronted with the possible adverse effects of unhealthy behaviors with enhanced self-affirmation, they become more aware of the negative consequences of their actions (or inactions), while also becoming painfully aware of the risk they pose to themselves. It appears that self-affirmation facilitates consideration of the longer-term implications of maintaining one’s current behavior when reading the health risk information, increases subsequent readiness for health behavior change, and gives individuals the necessary confidence and ability to engage in advocated health behaviors [22].

In intervention studies, self-affirmation commonly entails asking individuals to choose an important value and write about why and how they uphold it [23]. This is because people realize that their sense of self-integrity does not solely depend on the perceived implications of the provoking threat [24]. Specifically, affirming a significant aspect of oneself unrelated to the threatened domain may allow one to maintain or restore one’s overall, positive self-concept and adopt a longer-term perspective [25], which results in more open-mindedness and more adaptive responses to messages. Therefore, preceding health messages with self-affirmation interventions may increase intentions to engage in risk-reducing behaviors [15]. Notably, as acceptance of a risk message should reflect recognition of both the general risk of the behavior described in the message and the personal implications of engaging in the behavior [26], previous research also suggested manipulating both self-affirmation and health risk information within one experimental design [22].

However, most studies examined the effects of self-affirmation on cognition (e.g., attitudes) and proximal behavior (e.g., intentions) rather than on behaviors in the real world [27]. Additionally, although a meta-analysis discovered that self-affirmation manipulations had an average behavioral effect on responses to threatening information [28], the effects of self-affirmation are relatively modest and heterogeneous [29,30]. Moreover, fewer studies have examined how self-affirmation may exacerbate or buffer the relationship between negative self-evaluative emotions and healthy food choices [31,32,33,34,35]. Therefore, a secondary aim of the present study was to examine whether self-affirmation moderated the relationship between negative self-evaluative emotions and healthy food choices.

The current study hypothesized that greater negative self-evaluative emotions were associated with increased healthy food choices. We further hypothesized that self-affirmation would moderate the relationship between negative self-evaluative emotions and healthy food choices, and higher self-affirmation would exacerbate the impact of negative self-evaluative emotions on healthy food choices.

## 2. Materials and Methods

### 2.1. Participants

A total of 124 participants entered the survey system, but only 100 participants completed all the tasks. All the participants were Chinese. The final sample (N = 100) consisted of 60 (60%) female participants and 40 (40%) male participants, and the age ranged from 17 to 29 (*M* = 20.2 *SD* = 8.1). All the participants were classified as highly educated (97% bachelor students, 2% master students and 1% PhD level or above).

### 2.2. Recruitment and Procedure

One hundred and twenty-four student participants were recruited from Guangdong Medical University in China with a preregistration form on the official social media chat groups of the university. The form briefly mentioned that the research is about studying individuals’ dietary patterns, and the participants were asked to carefully read the explanations and then register for the next step (see registration form in the Appendix A). We used a depersonalized online recruitment system to recruit participants. It is worth mentioning that participation was completely voluntary, and participants were free to join our experiment or ignore these recruiting messages. Each subject was rewarded 0.5 course credits after completing the experiment.

Subjects who registered for our experiment successfully were invited to participate in the online experiment within one week. During the experiment, each subject was required to pass the attention check (i.e., can you please write twenty-two in Arabic numbers below?) and we put this attention check randomly. This check was used to test whether participants concentrated on participating in the experiment, especially in the online setting. If subjects fail the task, they have to restart the experiment. The experimental tasks include three main aspects, namely, questions about the manipulation of increasing self-affirmation, questions about the manipulation of stimulating negative self-evaluative emotions, and questions about food purchasing decisions. After completing the experimental tasks, subjects were asked to fill out a demographic survey. During this survey, we collected several demographic variables from subjects, including age, gender, monthly income, educational level, etc. Last but not least, we arranged a debriefing session to explain the research purpose of our study to participants via email, since we would like to let students not only get credits from participating in our experiment but also gain some knowledge about the research.

### 2.3. Design and Manipulations

The experiment was a 2 × 2 between-subject design and participants were randomly assigned to 4 different groups, with 25 participants per group. To be specific, participants of treatment group 1 received the intervention of stimulating negative self-evaluative emotions; participants of treatment group 2 received the intervention of increasing self-affirmation; participants of treatment group 3 received the intervention of stimulating negative self-evaluative emotions and increasing self-affirmation; participants of the control group, namely treatment group 4, neither received the intervention of stimulating negative self-evaluative emotions nor increasing self-affirmation (see more about online experiment in Appendix A).

For the manipulation of increasing self-affirmation [36], we asked participants first to indicate the domain they value the most and the domain they value a medium amount from a list of eight domains, including religion, politics, social aspects of life, economy, esthetic, theory, environment, and hedonism. After that, subjects of treatment groups 2 and 3 were provided five questions about the domain they had indicated to be the most important for them among these eight domains; in contrast, participants of treatment group 1 and the control group answered the five questions about their medium-important domain. The rationale of this manipulation procedure is that self-affirmation will occur when participants have the chance to define themselves clearly by repeatedly choosing the option that is related to their most important value [37]. It is worth mentioning that the method we use to increase self-affirmation is slightly different from previous research, such as the study of Dijkstra [11]. In that study, to decrease individuals’ self-affirmation, subjects who were from the groups without manipulation of increasing self-affirmation were asked questions about people’s least important value. Yet, exploring the impact of decreasing self-affirmation was not one of the purposes of our study, and we thus believe that in groups without increasing self-affirmation, asking people to answer questions about their medium-important value was appropriate.

For the manipulation of stimulating negative self-evaluative emotions, subjects of treatment groups 1 and 3 were asked to read a short message describing the potential consequence of unhealthy food consumption based on a person’s real experience (i.e., a young adult died due to unhealthy food eating). In contrast, subjects in treatment group 2 and the control group were asked to read a brief message about the same topic describing some normal health risk information (i.e., some influential data describing unhealthy food consumption could increase the risk of chronic disease). To check whether we manipulate negative self-evaluative emotions successfully, we asked three questions (i.e., (1): to what extent do you feel uncomfortable when reading the message; (2) to what extent do you feel stressed when reading the message; and (3) to what extent do you feel sorry for what the message described) to all subjects after the manipulation of negative self-evaluative emotions to compare whether individuals who receive treatment for increasing negative self-evaluative emotions will be more regretful than individuals who do not receive this treatment. These three questions aim to understand individuals’ extent of negative self-evaluative emotions from three different perspectives, namely individuals’ comfortability, stress, and guilt in reading this message. A Likert seven-point scale (1 = strongly disagree, 7 = strongly agree, so higher points imply more uncomfortable, stressful, or guilty) was used to measure the extent of regret for subjects since previous studies suggest that the human mind has a span of absolute judgment that can distinguish about seven distinct categories, a span of immediate memory for about seven items, and a span of attention that can encompass about six objects at a time [38]. The average of total points obtained by subjects who received the treatment for stimulating negative self-evaluative emotions was significantly higher than the average points of subjects who did not receive the treatment for increasing negative self-evaluative emotions. Both group 1 V.S. group 4 (t (48) = 4.87, *p* < 0.001) and group 2 V.S. group 3 (t (48) = −6.58, *p* < 0.001) showed significant differences (under 0.05 significance level). Additionally, ANOVA and T-test did not find that any factor (i.e., self-affirmation, income, age, gender, educational level) has a significant impact on the average points of the questions about the manipulation check. So, we can sufficiently demonstrate that the manipulation of negative self-evaluative emotions was successful and not confounded by other variables.

The questions about food purchasing decisions were designed as follows: each subject went through two rounds of food shopping and chose which food product they liked most. In each round, there were four options available, and two of them were unhealthy products (i.e., high sugar or high fat), while the rest of the two were healthy products (i.e., low or zero sugar or low fat). To avoid the influence of price differences, we made the price differences among these food products shown in each round as small as possible.

### 2.4. Measures

We aimed to measure the effect of stimulating negative self-evaluative emotions as the independent variable (IV). To achieve this, we utilized a dummy variable to capture the IV. Subjects who received treatment for increasing negative self-evaluative emotions were counted as 1 for their IV, and subjects who did not receive treatment for stimulating negative self-evaluative emotions were counted as 0 for their IV. The moderator we intended to measure was whether self-affirmation increased. Similar to our IV, we used a dummy variable to capture the moderator. Subjects who received the treatment for increasing self-affirmation were counted as 1 for their moderator, and subjects who did not receive treatment for increasing self-affirmation were counted as 0 for their moderator. The dependent variable (DV) we were interested in is healthy food consumption, measured by the number of healthy foods chosen by each subject. As mentioned above, each subject went through two rounds of purchasing, and they could pick one of the products in each round. Hence, the maximum quantities of healthy food that subjects could choose were 2, and the minimum quantities of healthy food that subjects could choose were 0. We compared the average number of healthy foods selected for participants in each group. A higher average number of healthy foods selected implies healthier diets.

### 2.5. Statistical Analysis

All statistical analyses were performed using SPSS 26 statistics and R (Version 4.3.1). Descriptive statistics for sociodemographic characteristics and the measure of healthy food consumption were presented separately by the four groups. One-way ANOVA (F-test) and Chi-squared test (χ^2^-test) were also conducted to examine the variable differences among four groups. Two-way ANOVA and moderation models were performed to examine the impact of negative self-evaluative emotions on healthy food choices, and the moderation effect of self-affirmation on such impact. Bootstrap analysis was used to test moderation effects, with a 95% confidence interval not including 0 indicating a significant mediation effect. A significance level of *p* < 0.05 was used to denote statistical significance.

## 3. Results

Descriptive analyses are displayed in Table 1. 

A two-way ANOVA was conducted that examined the effects of negative self-evaluative emotions and self-affirmation on healthy food choices, particularly testing the hypothesis that greater negative self-evaluative emotions are associated with increased healthy food choices. Our dependent variable, healthy food choice, was normally distributed for the groups formed by the combination of the levels of negative self-evaluative emotions and self-affirmation as assessed by the Skewness and Kurtosi test. There was homogeneity of variance between groups as assessed by Levene’s test for equality of error variances. Results revealed that negative self-evaluative emotions have a significant and positive impact on healthy food choices (F(1, 96) = 8.807, *p* = 0.004, see Figure 1). However, there is no significant interaction between the effects of self-affirmation and negative self-evaluative emotions on healthy food choices (F(1, 96) = 0.793, *p* = 0.376, see Figure 1). Further analysis, controlling for demographic variables including age, gender, education levels, and monthly income, yielded consistent results, no significant interaction was observed (F(1, 96) = 0.796, *p* = 0.375, see Figure 2), but there is still a significant and positive impact of the negative self-evaluative emotions on healthy food choices (F(1, 96) = 6.938, *p* = 0.01, see Figure 2).

Second, we used the moderation process analysis and the results of this approach described that the negative self-evaluative emotions explain 29 percent (R^2^ = 0.2897) of the variation in the health food choice. The interaction term of “increasing self-affirmation” times “increasing negative self-evaluative emotions slightly impacted the decrease in unhealthy food consumption, but the effect was non-significant (ΔR^2^ =0.008, F (1, 92) = 0.796, *p* = 0.375). Moreover, the bootstrap results were considered as well (95% CI: −0.4891–0.3242), and the results were consistent with the previous one. 

In sum, the above two methods both demonstrate that compared to the subjects who did not receive the intervention of stimulating negative self-evaluative emotions, the subjects who receive the intervention of negative self-evaluative emotions were significantly less willing to choose unhealthy food. However, the moderating effect of self-affirmation was rather weak and not significant, which means that the subjects who received the interventions of negative self-evaluative emotions and self-affirmation did not show any significant difference in unhealthy food consumption compared to the subjects who received the intervention of stimulating negative self-evaluative emotions.

## 4. Discussion

The present research aims to understand how the effectiveness of negative self-evaluative emotions in increasing healthy food choices can potentially be strengthened by increasing self-affirmation. While the results align with most previous studies in finding that increasing negative self-evaluative emotions is associated with a reduction in unhealthy food consumption, the interaction effect between increasing self-affirmation and stimulating negative self-evaluative emotions was not statistically significant. Moreover, it is worth mentioning that we do not find direct evidence that health risk information causes a high degree of defensiveness in individuals, as the impact of stimulating negative self-evaluative emotions on healthy food consumption is fairly significant. We thus assume that the defensive reactions may not be attributed to health risk information with stimulating negative self-evaluative emotions. Another potential reason is that defensiveness may occur when the negative self-evaluative emotions increase to a certain level. However, neither increasing self-affirmation itself nor increasing self-affirmation and stimulating negative self-evaluative emotions altogether shows a significant impact on healthy food consumption (*p* value > 0.1). 

The implications obtained from the findings are threefold. Regarding the scientific significance, it helps to better understand how to use health risk information with stimulating negative self-evaluative emotions to induce health-related behavioral change, especially when the goal is to increase healthy food consumption. More importantly, this is the first study to empirically demonstrate that increasing self-affirmation is not a very efficient way to reduce the individuals’ defensiveness caused by negative emotions to increase the effectiveness of negative self-evaluative emotions in preventing unhealthy diets. The potential reason is as follows: as mentioned previously, the purpose for increasing self-affirmation is to mitigate people’s defensive reactions when they receive health risk information. However, our research shows that such defensive reactions do not occur when participants read health risk information with stimulating negative self-evaluative emotions, so it is logical that increasing self-affirmation has a fragile effect.

Regarding the societal implication, policymakers and educators can take the research findings into account to better use negative self-evaluative emotions to prevent unhealthy food choices by resolving the issue of high defensiveness, especially in developing countries such as China. Policymakers and educators can share the experiences of some consumers who suffer from illnesses due to being addicted to unhealthy foods in public, thereby highlighting the importance of healthy food choices to other consumers. In addition to promoting healthy food consumption, our findings can also be implemented in other fields and for other purposes, for instance, increasing negative self-evaluative emotions to encourage students to improve their academic performances.

This study also contributes to the marketing field. Previous research argues that it is difficult to promote healthy food choices since the price of healthy food is normally higher than unhealthy food [39]. Our study gives extensive recommendations to marketing managers that it is wise to use the health risk information with negative self-evaluative emotions to promote the sales of healthy food products, and even other health goods such as fitness equipment, by emphasizing the risks of unhealthy lifestyles. Meanwhile, increasing self-affirmation is another promising tool that can be combined with negative self-evaluative emotions to reach the same goal. Although combining the increase in self-affirmation and negative self-evaluative emotions is rarely applied in the marketing area, increasing self-affirmation is a common marketing method for sellers to let consumers enjoy shopping with elevated feelings of well-being [31]. Nevertheless, it remains uncertain how this technique can be actually used in practice, for the following reason: it is unlikely for retailers to promote healthy products by explicitly mentioning the drawbacks of unhealthy products since they always have to balance the interest of various suppliers. For example, the manager of a supermarket definitely will not want to promote Coca-Cola Zero by highlighting the detriments of other unhealthy drinks such as Fanta, as this will harm their sales; hence, future research is advised to fully consider sellers’ situations and make this health risk information with negative self-evaluative emotions softer and more easily implemented.

It is worth noting that our research contains several limitations: first, the main limitation of the current study is that our sample size was relatively small, which may restrict the significance of the moderating effects of self-affirmation. We predict that increasing self-affirmation would have significantly moderated the impact of stimulating negative self-evaluative emotions on unhealthy food consumption if our sample size had been larger. Secondly, we have not measured individuals’ defensiveness for reading different types of health risk information nor comprehensively explored how increasing self-affirmation could decrease individuals’ defensiveness for receiving a message with negative self-evaluative emotions. We encourage future research to contribute towards a deeper understanding of this issue and achieve greater mastery of the impact of negative self-evaluative emotions and the use of increasing self-affirmation. Thirdly, similar to most previous research, our study focuses on using self-affirmation to induce behavioral change, but how best to implement self-affirmation needs to be carefully considered [40] in the future. For instance, in most studies about self-affirmation, people are asked to choose the domain they value most from a choice set. Yet, the impact of self-affirmations on health behavioral change could vary on the domains they choose because the values they are most focused on can be related or unrelated to health [41,42]. Fourth, our online experiment is highly hypothetical since participants are not monetarily incentivized and their real behaviors may not be fully captured; nonetheless, our use of an attention check ensures our participants participate in the experiment attentively and make their choices carefully. Future research should aim to better replicate this experiment in the field and observe whether the findings are consistent with our online experiment. Lastly, all of our participants are students and importantly, they are students from a medical university; it is, therefore, predictable that they have higher health literacy and are more likely to be influenced by such health-related messages than other groups of people. In other words, our results lack generalizability. Future studies could think about implementing our intervention in other populations and check its results accordingly.

## 5. Conclusions

Overall, there is no doubt that health risk information causes unhappy and stressful responses in people but making good use of this form of information benefits the promotion of health behaviors. Compared to the normal health risk information, prompting people to consider the negative consequences of their choices in advance and to become more regretful over their potential unhealthy behaviors seems to be a more promising solution for increasing healthy food choices. Importantly, more evidence is needed to demonstrate that self-affirmation is complementary to negative self-evaluative emotions in helping to push health behavioral change.

## Figures and Tables

**Figure 1 behavsci-14-00538-f001:**
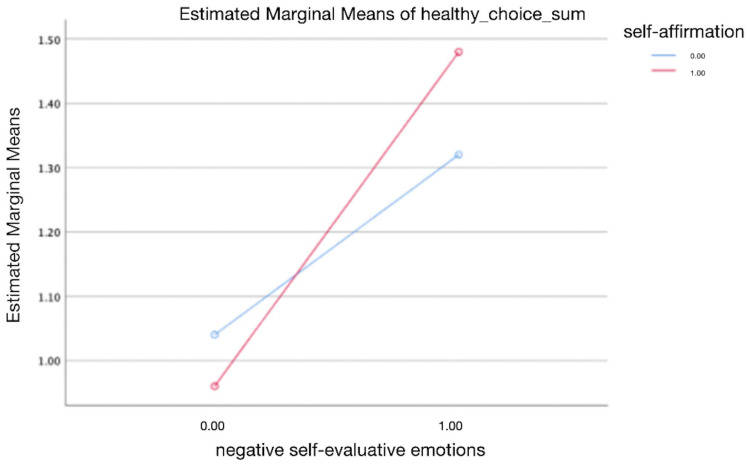
The change of unhealthy food consumption between two different situations of negative self-evaluative emotions with and without self-affirmation.

**Figure 2 behavsci-14-00538-f002:**
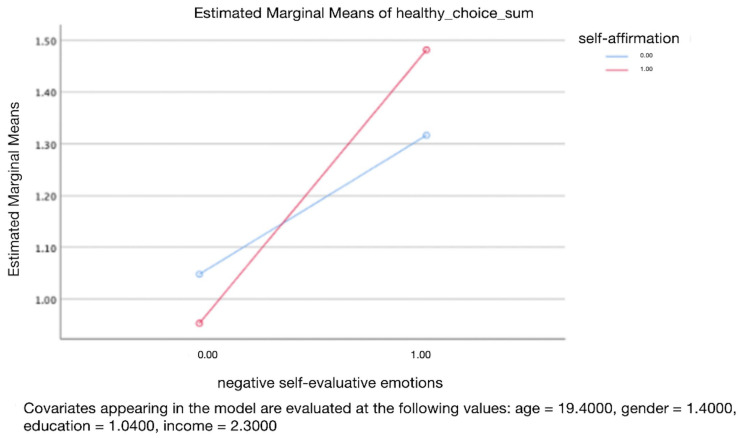
The change of unhealthy food consumption between two different situations of negative self-evaluative emotions with and without self-affirmation. Control variables: age, gender, education levels, and monthly income.

**Table 1 behavsci-14-00538-t001:** Descriptive information for participants and study variables.

Variables	Group				
	1 (NEG) (N = 25)	2 (AFF) (N = 25)	3 (NEG + AFF) (N = 25)	4 (CTL) (N = 25)	
Age (years, MEAN (SD))	19 (0.74)	19.2 (0.76)	18.9 (1.05)	20.6 (2.1)	F-test F = 9.26 *p* < 0.001
Gender					
Male	14	11	10	5	χ^2^ Test χ^2^ = 7 *p* = 0.072
Female	11	14	15	20	
Education					
Undergraduate	25	25	23	24	χ^2^ Test χ^2^ = 5.11 *p* = 0.529
Graduate	0	0	1	1	
Ph.D.	0	0	1	0	
Income (CNY)					
<1000	3	3	6	3	χ^2^ Test χ^2^ = 18.7 *p* = 0.227
1000–1500	9	18	12	15	
1500–2000	8	3	4	5	
2000–2500	5	1	2	1	
2500–3000	0	0	1	0	
>3000	0	0	0	1	
Healthy food choice round 1					
0	4	6	6	7	χ^2^ Test χ^2^ = 1.07 *p* = 0.784
1	21	19	19	18	
Healthy food choice round 2					
0	12	20	7	17	χ^2^ Test χ^2^ = 15.9 *p* = 0.001
1	13	5	18	8	
Healthy choice 2 rounds					
0	4	5	2	5	χ^2^ Test χ^2^ = 12.2 *p* = 0.057
1	9	16	9	14	
2	12	4	14	6	

Notes. Treatment group 1 (NEG): participants received an intervention designed to stimulate negative self-evaluative emotions. Treatment group 2 (AFF): participants received an intervention aimed at increasing self-affirmation. Treatment group 3 (NEG + AFF): participants received both interventions, aimed at stimulating negative self-evaluative emotions and increasing self-affirmation. Control group (CTL): denoted as treatment group four, participants did not receive either intervention (for more details on the online experiment, refer to the Appendix A). χ^2^ = squared chi. Healthy food choice round1, healthy food choice round2, and healthy choice 2 rounds are the number of healthy foods that participants chose.

## Data Availability

The data presented in this study are available on request from the corresponding author.

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
