# Peer review of "Do Negative Self-Evaluative Emotions Enhance Healthier Food Choices? Exploring the Moderating Role of Self-Affirmation"

_behavsci, 2024, doi:10.3390/bs14070538_

Round 1

Reviewer 1 Report

Comments and Suggestions for Authors

I believe that the authors have made the corrections and/or provided justifications satisfactorily and therefore I feel that the article is now suitable for publication. I have only two minor comments:

- In line 346, replace "F-test" by "one-way ANOVA (F-test)" and "Chi-square test" by "Chi-squared test (X2-test)".

- replace "X2 Tests" by "X2 test" in Table 1.

Note: X2 == squared chi (greek letter)

Author Response

Thank you very much for taking the time to review this manuscript. Please find the detailed responses below and the corrections highlighte in red in the re-submitted manuscript.

Reviewer 2 Report

Comments and Suggestions for Authors

In this study, authors conducted an online experiment with 100 student participants to examine if increasing self-affirmation can increase the effectiveness of negative self-evaluative emotions in reducing unhealthy food consumption. They found that negative self-evaluative emotions can significantly increase healthy food consumption. Self-affirmation was shown to have no moderating impact on the effectiveness of negative self-evaluative emotions in reducing unhealthy consumption. This study is well designed and well executed. The results are very precious. However, I found some parts that need revisions. I have some suggestions to improve clarity and accuracy. I hope that my comments are helpful for authors.

Line 20, “Self-affirmation was shown to have a positive moderating impact on the effectiveness of negative self-evaluative emotions in reducing unhealthy consumption, but not to a significant degree”. This expression seems misleading. Self-affirmation was found to have no moderating impact on the effectiveness of negative self-evaluative emotions in reducing unhealthy consumption.

Line 16, “intervention ineffectiveness could be weakened”. This expression is indirect.

Line 70, “The evidence for the intervention effectiveness of negative self-evaluative emotions for food choice among general populations remains scarce”. I agree that extending previous findings to under-studied populations is important. It is better to note why extending previous findings to general populations is important. Doing so is important since authors pointed out the importance of population-based promotion to reduce unhealthy food choices. General population is a targeted population for such promotions.

Line 103, “Dijkstra [22]”. This information is not consistent with that in the reference section (line 447).

Line 124, “Moreover, fewer studies have examined how self-affirmation may exacerbate or buffer the relationship between negative self-evaluative emotions and healthy food choices”. Readers want to know what these “few studies” have examined and clarified and what should be resolved. It is better to note such information.

2.3. Design and Manipulations, group one and group 1. It seems better to unify these different expressions.

Line 234, “The independent variable (IV) we wanted to measure was whether stimulating negative self-evaluative emotions”. Is this sentence incomplete?

Line 263, “A two-way ANOVA was conducted that examined the effects of negative self-evaluative emotions and self-affirmation on healthy food choices”. Are there any results about main effects of negative self-evaluative emotions and self-affirmation? Readers seem want to know if there is a significant main effect of negative self-evaluative emotion on healthy food choices. This is because examining this main effect is one of good tests for the hypothesis, greater negative self-evaluative emotions were associated with increased healthy food choices.

Line 301, “The results are in line with most previous studies in finding that increasing negative self-evaluative emotions is an effective way to prevent unhealthy food consumption, and that increasing self-affirmation slightly enhances the effectiveness of stimulating negative self-evaluative emotions in decreasing unhealthy food consumption”. A interaction effect is not significant. It seems better to note “The results are in line with most previous studies in finding that increasing negative self-evaluative emotions is an effective way to prevent unhealthy food consumption, but are not line with those in finding and that increasing self-affirmation slightly enhances the effectiveness of stimulating negative self-evaluative emotions in decreasing unhealthy food consumption”.

Author Response

Thank you very much for taking the time to review this manuscript. Please find the detailed responses below and the corrections highlighted in red in the re-submitted manuscript.

Round 2

Reviewer 2 Report

Comments and Suggestions for Authors

My previous comments have been thoroughly addressed. 

Author Response

Dear reviewer,

Many thanks for your feedback.

Kind regards,

Shuangmiao